# Small Molecules Temporarily Induce Neuronal Features in Adult Canine Dermal Fibroblasts

**DOI:** 10.3390/ijms242115804

**Published:** 2023-10-31

**Authors:** Kiyotaka Arai, Fumiyo Saito, Masashi Miyazaki, Haruto Kushige, Yayoi Izu, Noritaka Maeta, Kazuaki Yamazoe

**Affiliations:** 1Department of Veterinary Surgery, Faculty of Veterinary Medicine, Okayama University of Science, 1-3 Ikoi-no-oka, Imabari 794-8555, Japan; v18m131mm@ous.jp (M.M.); v19m067kh@ous.jp (H.K.); n-maeta@ous.ac.jp (N.M.); k-yamazoe@ous.ac.jp (K.Y.); 2Department of Toxicology, Faculty of Veterinary Medicine, Okayama University of Science, 1-3 Ikoi-no-oka, Imabari 794-8555, Japan; f-saito@ous.ac.jp; 3Department of Laboratory Animal Science, Faculty of Veterinary Medicine, Okayama University of Science, 1-3 Ikoi-no-oka, Imabari 794-8555, Japan; y-izu@ous.ac.jp

**Keywords:** small molecules, adult canine dermal fibroblasts, neuronal features, neuronal induction

## Abstract

Several methods have been developed to generate neurons from other cell types for performing regeneration therapy and in vitro studies of central nerve disease. Small molecules (SMs) can efficiently induce neuronal features in human and rodent fibroblasts without transgenes. Although canines have been used as a spontaneous disease model of human central nerve, efficient neuronal reprogramming method of canine cells have not been well established. We aimed to induce neuronal features in adult canine dermal fibroblasts (ACDFs) by SMs and assess the permanency of these changes. ACDFs treated with eight SMs developed a round-shaped cell body with branching processes and expressed neuronal proteins, including βIII-tubulin, microtubule-associated protein 2 (MAP2), and neurofilament-medium. Transcriptome profiling revealed the upregulation of neuron-related genes, such as *SNAP25* and *GRIA4*, and downregulation of fibroblast-related genes, such as *COL12A1* and *CCN5*. Calcium fluorescent imaging demonstrated an increase in intracellular Ca^2+^ concentration upon stimulation with glutamate and KCl. Although neuronal features were induced similarly in basement membrane extract droplet culture, they diminished after culturing without SMs or in vivo transplantation into an injured spinal cord. In conclusion, SMs temporarily induce neuronal features in ACDFs. However, the analysis of bottlenecks in the neuronal induction is crucial for optimizing the process.

## 1. Introduction

The isolation of large numbers of primary neurons from patients with neural disorders is currently challenging. Therefore, the development of easy and efficient methods to generate neurons from other cells can contribute considerably to the advancement of regenerative therapy and in vitro studies of central nerve disease. The development of methods to generate induced pluripotent stem (iPS) cells confers significant advantages to researchers [1,2]. However, processes involved in the generation, maintenance, and neuronal induction of iPS cells are expensive and time-consuming, which limits their application in regenerative therapy and experimentation. Recently, methods to convert fibroblasts into neurons using small molecules (SMs) without needing pluripotent cells or transgene techniques have been developed [3,4]. Some cocktails of SMs for neuronal induction of human fibroblasts have been reported, such as glycogen synthase kinase-3 (GSK-3) inhibitors, adenylyl cyclase activators, transforming growth factor β (TGFβ) inhibitor, and some inhibitors and activators that promote neuronal induction, survival, and maturation [5,6,7,8,9,10,11,12,13]. Histone deacetylase inhibitor can be concomitantly used with these SMs to modify the original epigenetic memory. Persistent treatment with these SMs activates intracellular signaling pathways that promote the expression of neuronal transcriptional factors, leading to neuronal fate. Furthermore, cytokines such as neurotrophins and fibroblast growth factor 2 (FGF2) have been used to support neuronal induction during SM treatment.

Similar to human cells, rodent fibroblasts have been successfully converted into neurons using SMs [10,11,12,13], suggesting the potential applicability of this method for the induction of neuronal features in fibroblasts of various species. Canines have often been used as spontaneous human disease models, such as spinal cord injury models, because their living environment and disease pathogenesis are similar to those of humans. However, an efficient neuronal reprogramming method of canine cells have not well established. In this study, we aimed to induce neuronal features in adult canine dermal fibroblasts (ACDFs) using SMs to assess the feasibility of SM-induced neuronal reprogramming for cell replacement and transplantation therapy. We conducted a morphological analysis, immunocytochemistry studies, transcriptome analysis, and calcium imaging to confirm the acquisition of neuronal features in ACDFs. Further investigation conducted through two-dimensional (2D) culture, basement membrane extract (BME) droplet 3D culture, and in vivo studies of spinal cord injury revealed a loss of induced neuronal features in ACDFs, suggesting the transience of neuronal induction. Therefore, our findings provide evidence that SMs can temporarily induce neuronal features in ACDFs.

## 2. Results

### 2.1. Induction of Neuronal Features in ACDFs

We used the tissue explant method to isolate fibroblasts [14]. This technique has been widely used in previous studies to avoid using enzymes that can contaminate other cells. Upon examination, most cells that migrated from the skin explants (Figure 1A) exhibited fibroblastic-spindle and flat forms (Figure 1B) and produced type I collagen (Figure 1C). In contrast, only a few cells expressed MAP2 and βIII-tubulin (Figure 1D). Neurofilament-medium (NF-M; Figure 1E) was completely negative in all ACDFs. To ensure the specificity of the antibodies used, we confirmed their ability to immunoreact with canine antigens (Appendix A).

To induce neuronal features in ACDFs, the effects of seven SMs that induce neuronal differentiation in human fibroblasts [6] were tested. After treating ACDFs with a cocktail of six SMs, a drastic change in morphology was noted, and they adopted neuronal morphology by Day 6 (Appendix A). The addition of Go6983, a protein kinase C (PKC) inhibitor, to the six SMs did not increase the proportion of neuronal morphology. However, according to reports that state Go6983 increases the expression of neuronal transcriptional factors in human fibroblasts [6], we included Go6983 in our protocol to upregulate neuronal genes. Next, we added Dorsomorphin, one of the bone morphogenetic protein (BMP) type I receptor inhibitors that are used in many protocols and contribute to induction of neuronal features in human and rat fibroblasts [5,7,8,12]. As a result, Dorsomorphin increased the proportion of neuronal morphology by 5% as compared to the seven SMs (Appendix A), which qualified Dorsomorphin for inclusion in our protocol. Next, we tested some combinations of cytokines for the application of neuronal induction medium (Appendix A). The addition of b-FGF with SMs increased the proportion of βIII-tubulin-positive cells and decreased nuclear fragmented dead cells (Appendix A). However, the further addition of neurotrophins did not improve induction efficiency and inhibit cell death. According to these results of preliminary study, ACDFs were cultured for 6 days in a neuronal induction medium containing eight SMs and b-FGF in the principal experiments. Maturation protocols were conducted after neuronal induction for 6 days as reported by Hu et al. [6]. The slightly modified protocol used in this study showed similar induction efficacy as the original protocol for human fibroblasts [6] on Day 12 (Appendix A).

Control samples cultured in a medium containing cytokines without SMs remained morphologically unchanged across all time points (Figure 1F–H,M). Conversely, SM-treated ACDFs underwent a drastic change in morphology and adopted a neuronal morphology characterized by round-shaped cell bodies with cell processes (Figure 1I–L). The proportions of ACDFs exhibiting neuronal morphology (Figure 1L) on Day 1 (80.9%), Day 6 (85.2%), and Day 12 (89.1%) were significantly increased compared to control cells across all time points (Figure 1M). Although they were not statistically significant, the proportions of ACDFs exhibiting neuronal morphology showed an increasing trend from Day 1 to Day 12. These findings suggest that SMs can induce neuronal morphology in ACDFs.

### 2.2. Exhibition of Neuronal Features in Small Molecule-Treated ACDFs

To assess the presence of neuronal features, control and SM-treated ACDFs were immunocytochemically analyzed on Day 6 and 12. In the vehicle-treated group, only a few cells were positive for MAP2 (Figure 2A,C) and βIII-tubulin (Figure 2B,C) and completely negative for NF-M (Figure 2D) on Day 6 and 12. Conversely, in the SM-treated groups, many cells showed positive immunoreactivity for MAP2 (Figure 2E,G), βIII-tubulin (Figure 2F,G), and NF-M (Figure 2H) on Day 6 and 12. The proportions of cells in the SM-treated group expressing MAP2 (35.4% and 41.6% on Day 6 and 12, respectively), βIII-tubulin (49.2% and 55.0% on Day 6 and 12, respectively), and NF-M (4.6% and 7.0% on Day 6 and 12, respectively) were significantly higher than those of the control group (Figure 2I). These findings suggest that the expression of neuronal proteins can be induced in ACDFs by SM treatment.

Next, for comprehensive analysis, we compared global gene expression profiles between ACDFs before and after 12-day SM treatment using microarray. Hierarchical clustering analysis revealed similar genome-wide changes between ACDFs before and after 12-day SM treatment (Figure 3A). A total of 15,170 genes were analyzed without duplication, where 1588 and 1456 genes were significantly upregulated and downregulated, respectively in SM-treated ACDFs as compared with ACDFs before SM treatment. Among the top 100 upregulated genes, 13 genes highly associated with neurons, such as *SNAP25*, *KCNA1*, and *GRIA4*, were identified (Figure 3B). Furthermore, among the top 100 downregulated genes, 10 genes highly associated with fibroblasts, such as *COL12A1*, *SVEP1*, and *CCN5*, were identified (Figure 3C). The gene ontology (GO) and pathway analyses revealed that neuron-related terms such as brain development, neuron projection development, and synaptic vesicle cycle were upregulated in SM-treated ACDFs as compared with ACDFs before SM treatment (Figure 3D). On the contrary, fibroblast-related terms, such as tissue morphogenesis, extracellular matrix organization, and response to wounding, were highly downregulated (Figure 3E). Other general neuron-related genes, such as *ASCL1*, *NEFM*, and *CACNA1B*, were significantly upregulated in SM-treated ACDFs. Conversely, *MAP2* and *TUBB3* showed low expression levels (Figure 3F). Therefore, we conducted time-course quantitative real-time reverse transcription-polymerase chain reaction (qRT-PCR) and observed that *MAP2* and *TUBB3* were upregulated during the induction process and decreased toward Day 12 (Figure 3G,H). These findings suggest that SM treatment, at least partially and temporarily, can induce neuronal genetic features in ACDFs.

To investigate neuronal activity, we performed calcium imaging to detect neuronal calcium ion flow in SM-treated ACDFs on Day 12. A spontaneous intracellular calcium transition was observed in SM-treated ACDFs (Figure 4A,B) but rarely in ACDFs or control cells (Figure 4B). The stimulation of SM-treated ACDFs with glutamate or KCl drastically increased intracellular calcium fluorescent intensity, unlike in ACDFs or control cells (Figure 4C). In SM-treated ACDFs, the fluorescent intensity peaked around 75 s after glutamate (Figure 4D) or KCl (Figure 4E) stimulation. The intensity at 75 s after glutamate or KCl stimulation was significantly higher in SM-treated ACDFs than in ACDFs and control cells (Figure 4F,G). These findings potentially suggest that SM-treated ACDFs exhibit neuronal physiological activity.

### 2.3. Additional Treatment in 2D Culture, 3D Culture, and In Vivo Conditions of Spinal Cord Injury Diminishes Neuronal Protein Expression in Small Molecule-Treated Adult Canine Dermal Fibroblasts

Genetic analysis revealed that upregulated neuronal genes such as *MAP2* and *TUBB3* showed a trend to be diminishing the expression level, implying that neuronal induction is temporary. To clarify the stability of induced neuronal features, SM-treated ACDFs were subjected to 2D and 3D culturing as well as to in vivo conditions of spinal cord injury (Figure 5A). First, SM-treated ACDFs (Figure 5B) were cultured in 2D culture using a neuronal medium without SMs for 14 days. Surprisingly, several SM-treated ACDFs reverted to their original fibroblastic-flat morphology (Figure 5C) and showed diminished expression of neuronal proteins such as MAP2 (Figure 5D), βIII-tubulin (Figure 5D), and NF-M (Figure 5F). The proportions of cells expressing MAP2, βIII-tubulin, and NF-M were 0.4%, 19.4%, and 0%, respectively, on Day 26. A previous study suggests that 3D-culture conditions may promote reprogramming efficacy [15]. Furthermore, it was reported that a brain-like microenvironment can facilitate neuronal maturation [16]. Thus, we hypothesized that modifying the microenvironment may help in maintaining neuronal phenotypes. To test this hypothesis, ACDFs were treated with SMs in 3D BME droplets. In line with 2D culture, most cells in the BME droplet-containing medium exhibited neuronal morphology and expressed MAP2 (Figure 5F,G), βIII-tubulin (Figure 5F,G), and NF-M (Figure 5H,I) on Day 12. The proportions of cells expressing MAP2, βIII-tubulin, and NF-M were 50.6%, 40.4%, and 4.8%, respectively (*n* = 4). However, the neuronal protein expression was drastically diminished after additional culture using a neuronal medium without SM for 14 days (Figure 5J–M). The proportions of cells expressing MAP2, βIII-tubulin, and NF-M were 0%, 1.7%, and 0%, respectively, on Day 26 (*n* = 4). Next, SM-treated ACDFs in the 3D BME droplets were implanted into a hemitransected rat spinal cord to examine the influence of the microenvironment of injured central nerve tissue on neuronal maintenance. Prior to embedding them in the BME droplets, ACDFs were labeled with aggregation-induced emission (AIE) dot fluorescent probes (Figure 5N,O). After 14 days of implantation, cells labeled with AIE dots were observed in the injured spinal cord (Figure 5P). However, most cells showed diminished expression of neuronal proteins, including MAP2 (Figure 5Q), βIII-tubulin (Figure 5R), and NF-M (Figure 5S). The proportions of cells expressing MAP2, βIII-tubulin, and NF-M were 1.4%, 0%, and 1.7%, respectively (*n* = 4), on Day 26. These findings suggest that the induction of neuronal features in ACDFs by SM treatment is temporary.

## 3. Discussion

In this study, we found that SM-treated ACDFs exhibited neuron-like characteristics; for instance, they possessed a round-shaped cell body and long cell processes, expressed neuronal protein, exhibited neuronal intracellular calcium ion flow, and upregulated neuronal gene expression, suggesting that SMs can induce neuronal features in ACDFs. These findings are generally consistent with the previous studies of human and rodent fibroblasts [5,6,7,8,9,10,11,12,13]. We used a combination of SMs, including a TGF-β inhibitor, GSK-3 inhibitor, histone deacetylase inhibitor, adenylyl cyclase activator, c-Jun N-terminal kinase (JNK) inhibitor, rho-associated coiled-coil-containing protein kinase inhibitor, BMP/AMP-activated protein kinase (AMPK) inhibitor, and PKC inhibitor to induce the transformation of ACDFs to neurons. RepSox (a TGF-β inhibitor) and CHIR99021 (a GSK-3 inhibitor) have been frequently used in various protocols to increase reprogramming efficacy in iPS cells, neuronal cells, and cardiomyocytes [3,4]. Valproic acid (a histone deacetylase inhibitor) can facilitate epigenetic reprogramming and neuronal differentiation [17,18]. In addition, Forskolin (an adenylyl cyclase activator) contributes to the efficient conversion of human fibroblasts into cholinergic neurons by inducing the expression of the transcription factor *NGN2* [19]. Similarly, SP600125 (a JNK inhibitor) promotes the reprogramming of human fibroblasts into neuronal stem cells by mediating the expression of the transcription factor *OCT4* [20]. Furthermore, Y27632 (a rho-associated coiled-coil-containing protein kinase inhibitor) can promote the differentiation of somatic stem cells into neurons [21]. These SMs have been widely employed in previous studies investigating SM-induced neuronal reprogramming in human and rodent cells [6,7,8,9,12,13]. In this study, BMP/AMPK and PKC inhibitors were employed for the neuronal induction process. Dorsomorphin, a BMP/AMPK inhibitor that promotes the transdifferentiation of T cells into neurons by transgenes [22], was used in this study. Additionally, Go6983 (a PKC inhibitor) can facilitate the differentiation of Neuro-2a cells into neurons [23]. Complex effects of these SMs may have contributed to the induction of neuronal intracellular signaling, resulting in the acquisition of neuronal features in ACDFs. In this study, when the neuronal induction process using eight SMs was completed on Day 6, the proportion of ACDFs showing neuronal morphology and expressing βIII-tubulin were 85% and 49%, respectively. These results indicate that some of ACDFs are resistant to SM-induced neuronal conversion. Further optimization of the induction process is necessary to achieve highly efficient conversion of ACDFs to neurons. This requires efficient and effective SM screening based on the neuronal induction mechanisms and reprogramming barriers.

The expression of neurofilament, a complex mature neuron protein, in SM-treated fibroblasts has not been thoroughly assessed thus far in human and rodent cells. In this study, we observed that the expression level of NF-M was low even after the neuronal maturation process, indicating that the maturation process is insufficient for ACDFs. We used a cocktail of three SMs during the maturation process: CHIR99021 (a GSK-3 inhibitor), forskolin (an adenylyl cyclase activator), and dorsomorphin (a BMP/AMPK inhibitor). Previous research has demonstrated that GSK-3 inhibition promotes neuronal differentiation in human neuronal progenitor cells [24]. Additionally, the activation of cAMP induces morphological maturation of neurons in the hippocampus [25]. Therefore, GSK-3 inhibitors and cAMP activators have been widely used in previous studies for neuronal maturation protocols [6,7,9,12,13] and were used in this study. It is worth noting that the use of a GSK-3 inhibitor may induce AMPK activation, which can suppress axon formation by inhibiting the mammalian target of the rapamycin signaling pathway [26]. This can be attributed to the GSK-3-mediated inhibition of AMPK activity through phosphorylation of the alpha subunit at Thr479 [27]. Therefore, the use of dorsomorphin as an AMPK inhibitor was considered appropriate. Although SP600125, a JNK inhibitor, has been used in conjunction with GSK-3 inhibitors and cAMP activators [12], we did not employ a JNK inhibitor as it can arrest neurofilament expression during neuronal differentiation of mouse embryonic stem cells [28] and reduce valproic acid-induced neurite outgrowth in mouse neural stem cell-derived neurons [29]. Valproic acid has been found to activate JNK signaling, resulting in the expression of neuronal markers [29]. Therefore, the use of valproic acid for JNK activation may be appropriate in the maturation protocol. Nevertheless, a more comprehensive understanding of the underlying mechanisms of neuronal maturation is required to optimize protocols and achieve the generation of fully matured neuronal cells.

The fate of neuron-like cells generated by SMs has not been thoroughly assessed thus far. Studies showed that the upregulated βIII-tubulin in human fibroblast tends to be decreased after or during neuronal induction by SMs [8,9]. In our study, we demonstrated that SM-treated ACDFs reverted to a fibroblastic-flat or spindle-shaped morphology and exhibited decreased neuronal protein expression in three different microenvironments including 2D and 3D culture and an in vivo condition of spinal cord injury. These findings indicate that the induction of neuronal features in ACDFs is temporary regardless of microenvironment. It is important to confirm the permanency of the induced neuronal features without induction factors, in order to confirm that the neuronal conversion was completely successful. Similar observations have been made in iPS cell generation, where interrupted reprogramming can result in a reversion to the original phenotype [30]. Partial iPS cell reprogramming of mature somatic cells can produce rejuvenated cells that maintain their original phenotype [31]. These studies imply that incomplete neuronal induction by SMs, which fails to fully reprogram the original epigenetic memory of ACDFs, results in a reversion to the original phenotype. Therefore, a comprehensive epigenetic analysis and screening for SMs that can effectively promote epigenetic reprogramming are necessary to completely generate permanently converted neurons from ACDFs. 

We first demonstrated that SMs induce neuronal features in 3D reconstructed BME in ACDFs. Previous studies demonstrate that SM-soaked beads [11] and SM-soaked hydrogel [13] embedded in rodent skin induced βIII-tubulin protein expression in skin cells. In addition, a porous silk 3D scaffold promoted SM-induced neuronal gene expression in rat fibroblasts [12]. Although the induction protocol is amenable to further improvements and adaptations for canines, these studies together with our finding suggest that SM-induced neuronal conversion could be used in various methods for regenerative therapy.

A limitation of this study is that all fibroblasts were isolated from younger canines (six canines are under 2 years old; one canine is 3 years old; two canines are 5 years old). A previous study observed decreased neuronal induction efficacy with increasing age [8]. Therefore, cells originating from older canines may show lower induction efficacy. For a future application in regeneration therapy, further research on its age-dependency is warranted. In addition, in order to use the neuronal induced cells for regenerative medicine, it is necessary to prove in vitro and in vivo that the cells have neuronal function to test the therapeutic potential. In this study, we have shown that calcium ion influx occurs in SM-treated ACDFs spontaneously and in response to glutamate or KCl stimulation. However, additional electrophysiological evaluation will more reliably assess neuronal function.

When conducting research in the anticipation of future applications in regenerative medicine, it may be appropriate to use canines rather than other species. Canines share not only the same physiological functions and aging phenomena, but also the same living environment as humans. Therefore, the two species share many common diseases and pathogenic processes. For instance, brain trauma and spinal cord injury in canines, like in humans, occur in traffic accidents and fall accidents. It is expected that clinical research and veterinary clinical trials using canine cells will be useful as reliable preclinical data to initiate clinical trials on humans.

In conclusion, our study suggested that SMs can temporarily induce neuronal features in ACDFs. However, a better understanding of the obstacles in generating neurons from ACDFs is critical for optimizing protocols and achieving high conversion efficacy, sufficient maturity, and permanent maintenance of the neuronal phenotype.

## 4. Materials and Methods

### 4.1. Isolation of ACDFs

ACDFs were isolated from nine beagle dogs using a skin tissue explant culture method [14]. Abdominal skin tissue samples were collected from three male and six female beagle dogs, weighing from 7.9 kg to 9.5 kg and ranging in age from 1 year and 5 months to 5 years and 8 months. All experimental procedures were approved by the Institutional Animal Care and Use Committee of Okayama University of Science (approval code numbers 2020-099: 17 September 2020 and 2021-136: 16 November 2021). The collected skin samples were carefully dissected to remove the subcutaneous fat and then cut into 5 mm square pieces. The tissue pieces were then placed into 0.1% gelatin-coated 6-well plates. They were cultured in fibroblast medium composed of Dulbecco’s Modified Eagle Medium (DMEM; Sigma-Aldrich, St. Louis, MO, USA) supplemented with 20% fetal bovine serum (FBS; Biowest, Riverside, MO, USA) and 1% antibiotics/antimycotics (Nacalai Tesque, Kyoto, Japan) and maintained under 5% CO_2_ at 37 °C. The tissue pieces were removed once the outgrowth and expansion of ACDFs were observed. Subsequently, ACDFs were passaged when they reached 80% confluency and cultured in DMEM containing 10% FBS and 1% antibiotics/antimycotics. ACDFs at passages 4–7 were used for the subsequent experiments conducted in this study.

### 4.2. Induction of Neuronal Features

The neuronal induction and maturation protocols were conducted following the methods described by Hu et al. [6] with minor modifications. ACDFs were plated on gelatin-coated 15 mm coverslips (Nacalai Tesque) at a density of 20,000 cells/cm^2^ and cultured in DMEM containing 10% FBS and 1% antibiotics/antimycotics under 5% CO_2_ at 37 °C. After 1 day, the cells were washed with phosphate-buffered saline (PBS) and then cultured in a neuronal induction medium for 6 days. The neuronal induction medium consisted of DMEM/F12 (Nacalai Tesque) and Neurobasal Medium (Thermo Fisher Scientific, Waltham, MA, USA) at a 1:1 ratio, 1% B-27 Supplement (Thermo Fisher Scientific), 0.5% N_2_ Supplement (Wako, Osaka, Japan), 100 μM dibutyryl cyclic adenosine monophosphate (dibutyryl cAMP; Sigma-Aldrich), 20 ng/mL basic FGF (b-FGF; Peprotech, Cranbury, NJ, USA), 1% antibiotics/antimycotics, and eight SMs: 3 μM CHIR99021 (MedChemExpress, Monmouth Junction, NJ, USA), 1 μM dorsomorphin (Cayman Chemical, Ann Arbor, MI, USA), 10 μM Forskolin (Cayman Chemical), 5 μM Go6983 (Cayman Chemical), 1 μM RepSox (MedChemExpress), 10 μM SP600125 (MedChemExpress), 5 μM Y-27632 (MedChemExpress), and 0.5 mM valproic acid (Cayman Chemical). Valproic acid was dissolved in H_2_O, while the other SMs were dissolved in dimethyl sulfoxide (Nacalai Tesque). On Day 3, half of the medium was replaced to new one, and on Day 6, the medium was changed to neuronal maturation medium. The neuronal maturation medium comprised DMEM/F12 and Neurobasal Medium at a 1:1 ratio, 1% B-27 Supplement, 0.5% N_2_ Supplement, 100 μM dibutyryl cAMP, 20 ng/mL b-FGF, 20 ng/mL brain-derived neurotrophic factor (BDNF, Peprotech), 20 ng/mL glial cell line-derived neurotrophic factor (GDNF, Peprotech), 20 ng/mL neurotrophin-3 (NT-3, Peprotech), 1% antibiotics/antimycotics, and three SMs: 3 μM CHIR99021, 10 μM Forskolin, and 1 μM dorsomorphine. Half of the medium was replaced every other day until Day 12, for 6 days. As controls, cells were cultured in the same medium using an equal volume of vehicles, but without the addition of SMs. For additional 2D and 3D cultures after SM treatment, cells were cultured without SMs in neuronal medium, which consisted of DMEM/F12 and Neurobasal Medium at a 1:1 ratio, 1% B-27 Supplement, 0.5% N_2_ Supplement, 100 μM dibutyryl cAMP, 20 ng/mL BDNF, 20 ng/mL GDNF, 20 ng/mL NT-3, and 1% antibiotics/antimycotics, until Day 26 for 14 days.

### 4.3. Morphological Analysis

To assess morphological changes in the induced neurons, high-magnification images (×100) were randomly acquired from each sample on Day 1, 6, and 12 using a microscope (CKX53, Olympus, Tokyo, Japan). A minimum of 50 cells were evaluated. Cells showing a round-shaped cell body and generating cell processes were considered to have neuronal morphology [32].

### 4.4. Immunocytochemical Analysis

The cells cultured on coverslips were washed with PBS and then fixed with 4% paraformaldehyde (Nacalai Tesque) for 10 min at 20–25 °C. After fixation, the cells were treated with 5% goat serum (Vector Laboratories, Burlingame, CA, USA) and 0.2% Triton X-100 (Nacalai Tesque) for 30 min at 20–25 °C and then incubated for 18 h at 4 °C with primary antibodies diluted in PBS. The primary antibodies used in this analysis included mouse antibodies against type I collagen (1:300, Abcam, Cambridge, England), βIII-tubulin (1:300, Abcam), neurofilament-medium (NF-M; 1:300, Abcam), and rabbit antibodies against MAP2 (1:300, Abcam). After incubation with primary antibodies, the cells were washed with PBS and then incubated with the appropriate secondary goat antibodies diluted in PBS for 45 min at 20–25 °C. The secondary antibodies used were anti-mouse IgG conjugated with Alexa Fluor 594 (1:500, Abcam) or anti-rabbit IgG conjugated with Alexa Fluor 488 (1:500, Abcam), and anti-rabbit IgG conjugated with Alexa Fluor 594 (1:500, Abcam). A water-soluble mounting medium containing 4′,6-diamidino-2-phenylindole (DAPI; SouthernBiotech, Birmingham, AL, USA) was used to stain nuclei. To validate the specificity of the antibodies used in this study for canine antigens, 4% paraformaldehyde-fixed frozen sections of canine spinal cord tissue were employed. This tissue was collected from a euthanized canine after an autopsy conducted for an unrelated study. Cells incubated with mouse IgG_1_ isotype control (BioLegend; San Diego, CA, USA), IgG2a isotype control (BioLegend), or rabbit IgG isotype control (GeneTex; Irvine, CA, USA) at the same concentrations as the primary antibodies were used as negative controls. To calculate the proportion of immunopositive cells, high-magnification images (×200) were randomly acquired from each sample using an LSM880 confocal laser scanning microscope (Carl Zeiss, Oberkochen, Germany). A minimum of 50 cells identified with DAPI were analyzed using Fiji ImageJ software version 2.0.0 [33]. Original images were processed into binary images to identify fluorescent positive cells clearly. Cells showing high fluorescence signaling throughout cell body and processes were counted as positive. Cells low-signaling or showing partial and minute dot signaling were considered as negative. Dead cells were identified based on morphological changes observed in the nuclei stained with DAPI, with those displaying nuclear condensation or fragmentation considered dead [34].

### 4.5. Microarray

A total of eight RNA samples were extracted from ACDFs isolated from four different canines, both before and after a 12-day treatment with SMs, using the NucleoSpin RNA Kit (Takara Bio, Shiga, Japan) according to the manufacturer’s instructions. The RNA quantity was measured using the NanoDrop OneC spectrophotometer (Thermo Fisher Scientific), and the RNA quality was assessed using the TapeStation4150 system (Agilent Technologies, Santa Clara, CA, USA), which yielded high-quality results (RNA integrity number 9.7–10.0/10.0). Transcriptome profiling was conducted using the Canine (V2) Gene Expression Microarray (Agilent Technologies). Total RNA was amplified and labeled separately with Cy3 dyes, followed by hybridization onto slides. The microarray slides were scanned using a G2565CA Microarray Scanner (Agilent Technologies). The resulting array images were analyzed using Agilent Feature Extraction software (v10.7) (Agilent Technologies), and the expression data were further analyzed using Subio Platform software (Subio Inc., Kagoshima, Japan). Hierarchical clustering and heat mapping of the gene expression data were conducted using Heatmapper [35]. Additionally, GO and Kyoto Encyclopedia of Genes and Genomes pathway enrichment analyses were performed using Metascape [36].

### 4.6. qRT-PCR

A total of 16 RNA samples were extracted from ACDFs isolated from four different canines, before and after 3-, 6-, and 12-day treatment with SMs, using the NucleoSpin RNA Kit (Takara Bio) according to manufacturer’s instructions. RNA quantity was measured using a NanoDrop OneC spectrophotometer (Thermo Fisher Scientific). Complementary DNA was synthesized from total RNA using ReverTra Ace qPCR RT Master Mix with a gDNA remover (TOYOBO, Osaka, Japan) as per manufacturer’s instructions. Quantitative RT-PCR was performed in QuantStudio5 (Thermo Fisher Scientific) with THUNDERBIRD Next STBR qPCR Mix (TOYOBO) and the following settings: one cycle of denaturation at 94 °C for 30 s, followed by 40 cycles of denaturation at 95 °C for 5 s and annealing/extension at 60 °C for 20 s. The primer sequences used for qPCR were as follows: MAP2 [Sequences (5′->3′): Forward; CTTGACAATGCCCACCATGTAC, Reverse; TGCCTGGGGACTGTGTAATG, Product length: 127 bps], TUBB3 [Sequences (5′->3′): Forward; TTGAGCCCGGAACCATGG, Reverse; ACCTTTGGCCCAATTGTTGC, Product length: 110 bps], GAPDH [Sequences (5′->3′): Forward; TGTTTGTGATGGGCGTGAAC, Reverse; TGGATGACTTTGGCTAGAGGAG, Product length: 106 bps]. GAPDH was used as internal control to normalize the data. The relative gene expression levels were calculated using the standard curve method.

### 4.7. Ca^2+^ Imaging

Ca^2+^ imaging of cells cultured in a 96-well plate was performed using a Calcium Kit-Fluo4 (Dojin, Kumamoto, Japan) according to the manufacturer’s instructions. After the cells were washed with PBS, they were incubated in a loading buffer containing 3 μM Fluo4-AM, 0.04% Pluronic F-127, and 1.25 mM Probenecid at 37 °C for 45 min. Subsequently, the cells were washed again with PBS and transferred to a recording medium containing 1.25 mM Probenecid. Measurements were conducted using the green fluorescent protein channel, and fluorescent imaging was captured using BZ-X810 All-in-One Fluorescent Microscopy (Keyence, Osaka, Japan) under 5% CO_2_ at 37 °C. A movie sequence was recorded, capturing one image every 3 s. After observing for 60 s, the cells were stimulated with 100 μM glutamate (Nacalai Tesque) or 50 mM KCl (Nacalai Tesque), and images were captured for 150 s after stimulation. The fluorescence intensity was analyzed using Fiji ImageJ software [16]. After subtracting the background signal, the normalized fluorescent intensity was calculated using the formula ΔF/F0, where ΔF = F_time_ − F_0_. The fluorescence intensity of 60 cells (20 cells from three different ACDF lines) at 75 s after glutamate or KCl stimulation was compared between ACDFs, control ACDFs, and SM-treated ACDFs.

### 4.8. Induction of Neuronal Features in BME Droplets

BME (Cultrex path clear basement membrane extract, R&D Systems, Minneapolis, MN, USA) and neuronal induction medium (×3) were mixed at a 2:1 ratio. The cells were then dissolved in the mixture at a concentration of 2 × 10^4^ cells/μL at 4 °C. Next, the BME with cells was dropped onto a 24-well culture plate in 5 μL volumes and allowed to gel at 37 °C for 30 min. Once the gel formed, the BME droplets were cultured in a neuronal induction medium for 6 days, followed by a neuronal maturation medium for another 6 days. For the control cells, the same procedure was followed, but no SMs were added. The cultured BME droplets were fixed with 4% paraformaldehyde for 30 min at 20–25 °C and then cryoprotected with 20% sucrose. The specimens were cut into three 10-μm-thick sections using a cryostat. The immunostaining procedures were performed as per the same protocol for immunocytochemistry.

### 4.9. Animals and Their Experimental Procedures

The animal experiments were conducted with the approval of the Institutional Animal Care and Use Committee at Okayama University of Science (approval code number 2022-033: 9 April 2021). Rats were housed in a controlled environment with a 12/12 h light/dark cycle and a temperature of 26 °C. The rat spinal cord hemisection procedure was performed as described previously [37]. Laminectomy was performed at the thoracic segment nine of four healthy male Wistar rats (SLC, Shizuoka, Japan) that were nine weeks old and weighed 220–230 g. The procedure was conducted under anesthesia with intraperitoneal administration of a mixture of 0.15 mg/kg medetomidine (Meiji Seika Pharma, Tokyo, Japan), 2 mg/kg midazolam (Sandoz, Tokyo, Japan), and 2.5 mg/kg butorphanol (Meiji Seika Pharma). The middle of the exposed spinal cord was pierced using a 29-gauge needle, followed by cutting from the needle hole to the right lateral edge of the spinal cord using a No. 15 scalpel blade. Lastly, the complete hemitransection of the spinal cord was confirmed using Dumont forceps No. 5. An antibiotic (25 mg/kg cefazoline, LTL Pharma, Tokyo, Japan) and analgesic (5 mg/kg carprofen, Zoetis, Parsippany, NJ, USA) were subcutaneously injected twice a day and once a day, respectively, for 3 days. After 7 days of hemitransection, the hemitransected spinal cords of the rats were re-exposed under anesthesia, and SM-treated ACDFs in BME droplets were implanted. Before dissolving the ACDFs in the BME droplets, the cells were labeled with photostable fluorescent organic dots with AIE dots (LuminiCell Tracker 540-Cell Labeling Kit, Sigma-Aldrich) according to the manufacturer’s instructions for tracing cells [38]. A daily injection of immunosuppressant (10 mg/kg cyclosporine, Novartis Pharma, Basel, Switzerland) was administered to avoid rejection one day before surgery until euthanasia. An antibiotic (25 mg/kg cefazoline) and analgesic (0.3 mg/kg butorphanol) were injected subcutaneously twice a day for 3 days. After 14 days of implantation, rats were euthanized under deep anesthesia with isoflurane. After confirming respiratory arrest for 2 min, rats were transcardially perfused with 50 mL of saline, followed by 50 mL of 4% paraformaldehyde. The hemitransected spinal cords were carefully resected from each rat and post-fixed with 4% paraformaldehyde for 4 h. After incubation in 20% sucrose for cryoprotection, the specimens were cut into three 10 μm thick sections using a cryostat. The immunohistochemical procedures were performed using the same protocol as immunocytochemistry.

### 4.10. Statistical Analysis

The extent of morphological change and immunoreactivity of neuronal markers was compared between the SM- and vehicle-treated ACDFs using either a two-sample *t*-test or Mann–Whitney *U* test. For microarray data, a paired *t*-test was used to compare gene expression between pre- and post-treatment with SMs. For experiments involving more than three groups, data were compared using a one-way analysis of variance followed by Tukey’s test or the Kruskal–Wallis test. All data were presented as means and standard deviations. Statistical analyses were performed using EZR [39]. A *p*-value of less than 0.05 was considered statistically significant.

## Figures and Tables

**Figure 1 ijms-24-15804-f001:**
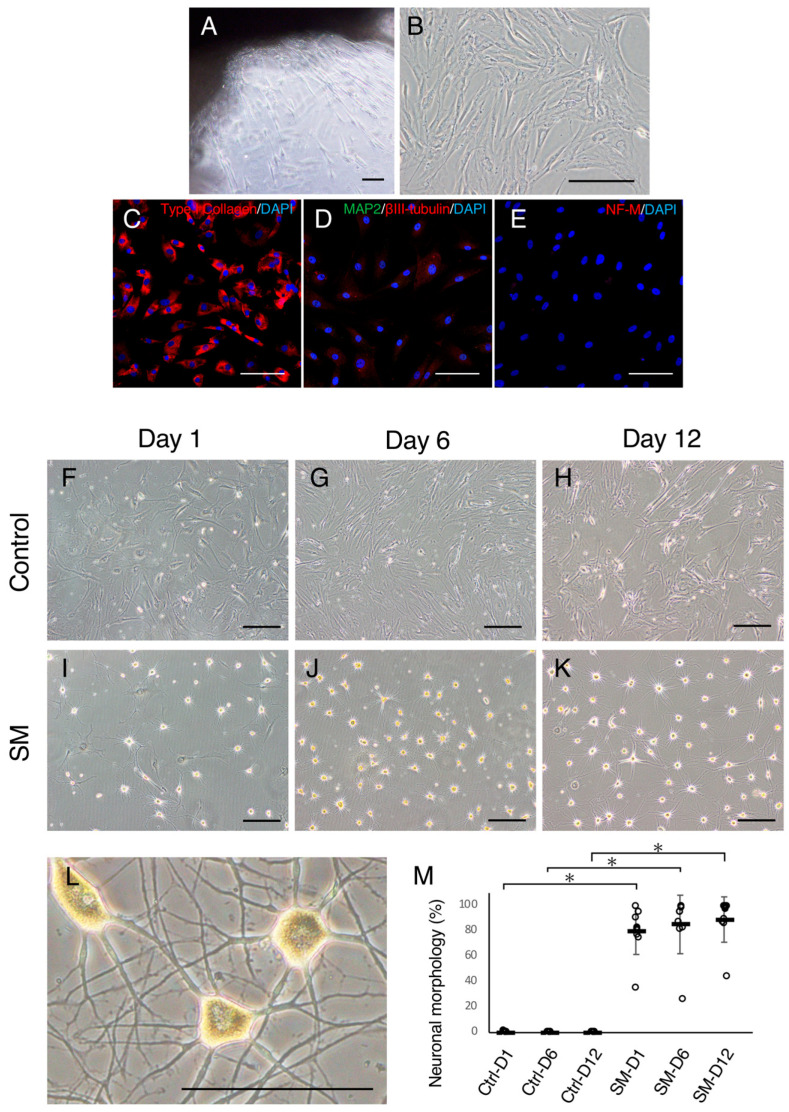
Assessment of isolated adult canine dermal fibroblasts (ACDFs) and induction of neuronal morphology in ACDFs. (**A**) Photomicrograph showing ACDFs growing outward from the skin explant and (**B**) expanding in the fibroblast medium. (**C**–**E**) Immunocytochemistry images of ACDFs for (**C**) Type I collagen, (**D**) βIII-tubulin/MAP2, and (**E**) neurofilament-medium (NF-M). (**F**–**K**) Morphological changes in ACDFs treated with vehicle control on (**F**) Day 1, (**G**) Day 6, and (**H**) Day 12 and selected small molecules (SMs) on (**I**) Day 1, (**J**) Day 6, and (**K**) Day 12. (**L**) A high-magnification image of ACDFs with neuronal morphology, possessing a round-shaped cell body and long cell processes. Bars = 400 μm in panel A, 200 μm in (**B**,**F**–**K**), and 100 μm in (**C**–**E**,**L**). (**M**) Graph presenting proportions of cells exhibiting neuronal morphology. White dots represent the proportion of cells exhibiting neuronal morphology in each of the canines. Black bars and error bars indicate the mean and standard deviations respectively (*n* = 9). * *p* < 0.001.

**Figure 2 ijms-24-15804-f002:**
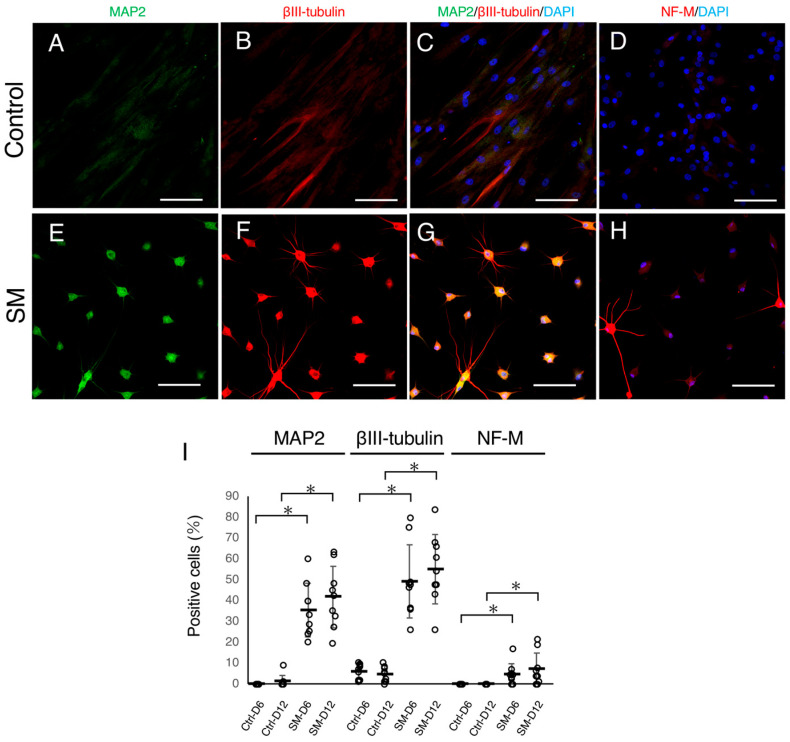
Neuronal protein expression in small molecule (SM)-treated adult canine dermal fibroblasts (ACDFs). (**A**–**H**) Immunocytochemistry images of (**A**,**E**) MAP2, (**B**,**F**) βIII-tubulin, and (**D**,**H**) neurofilament-medium (NF-M) in ACDFs treated with vehicle (control) or SMs on Day 12. Bars = 100 µm. (**I**) Graph presenting proportions of cells positive for MAP2, βIII-tubulin, and NF-M. White dots in the graph present the proportions of positive cells in each of the canines. Black bars and error bars represent the means and standard deviations, respectively (*n* = 9). * *p* < 0.001.

**Figure 3 ijms-24-15804-f003:**
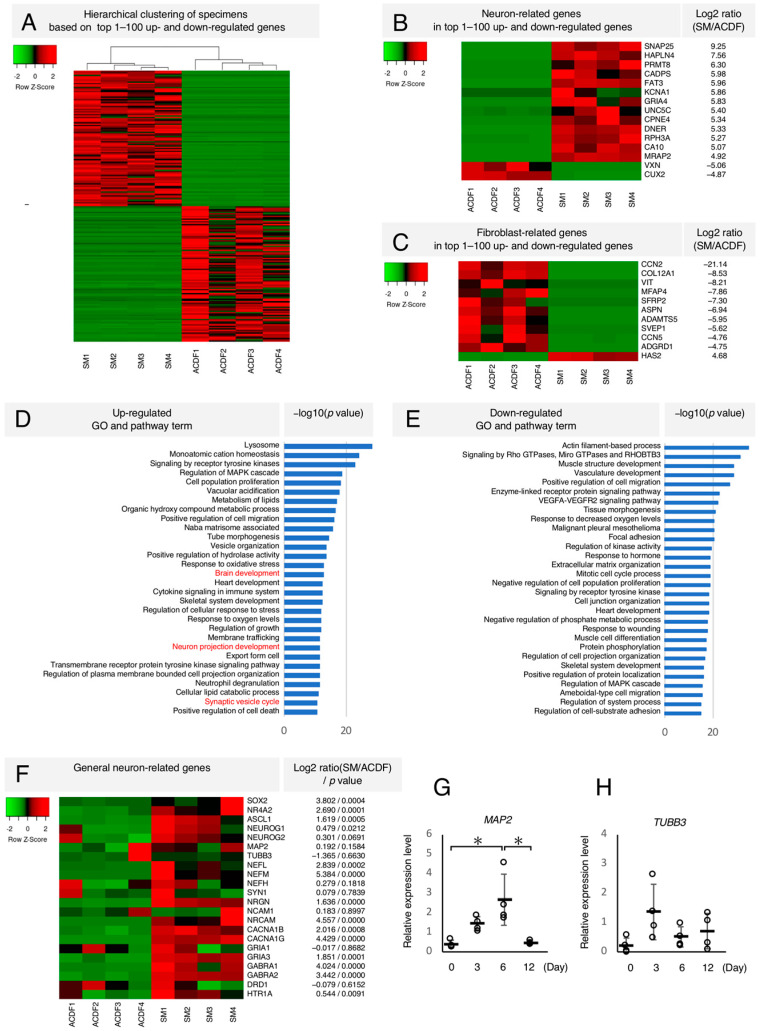
Microarray gene expression analysis. Changes in gene expression in adult canine dermal fibroblasts (ACDFs) before and after treatment with small molecules (SMs) (*n* = 4). (**A**) Hierarchical clustering representing the similarity of gene expression changes between ACDFs. (**B**) Heatmap of neuron-related genes from the top 100 upregulated and downregulated genes. (**C**) Heatmap of fibroblast-related genes from the top 100 upregulated and downregulated genes. (**D**,**E**) The graph represents highly (**D**) upregulated and (**E**) downregulated gene ontology (GO) and pathway terms in SM-treated ACDF. Neuron-related terms are represented in red. (**F**) Heatmap of general neuron-related genes. (**G**,**H**) Time-course analysis of relative gene expression level of (**G**) *MAP2* and (**H**) *TUBB3* using quantitative real-time reverse transcription-polymerase chain reaction. White dots represent the relative expression level of normalized genes in each of the canines. Black bars and error bars represent the means and standard deviations, respectively (*n* = 4). * *p* < 0.01.

**Figure 4 ijms-24-15804-f004:**
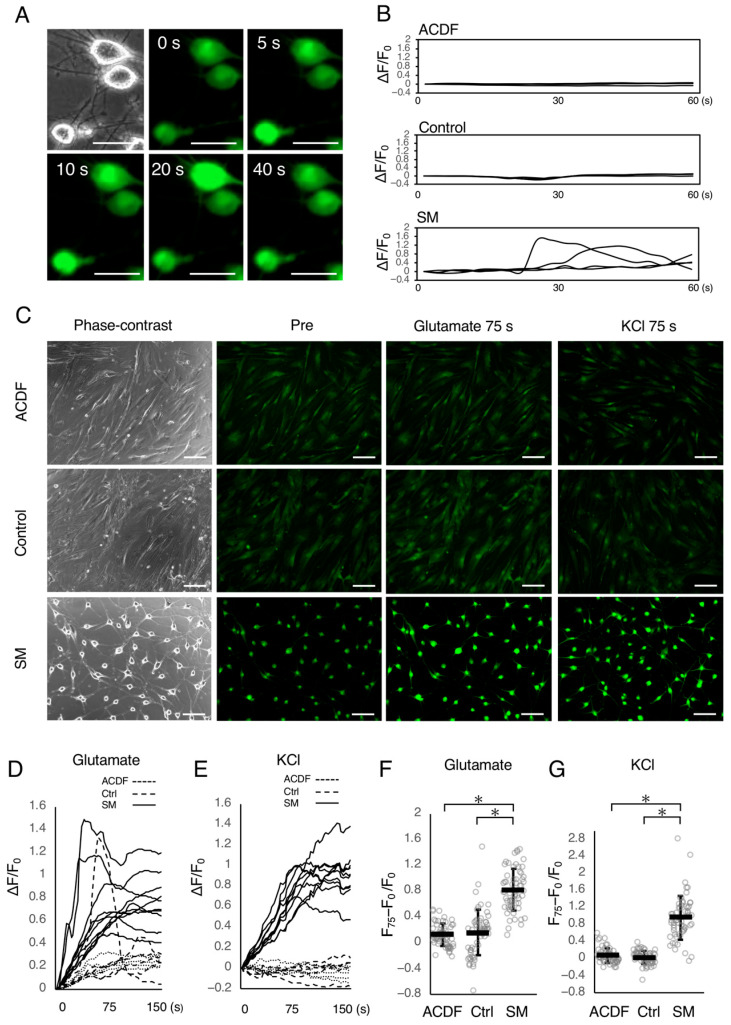
Stimulation-dependent intracellular Ca^2+^ response with Fluo-4 AM. (**A**) Fluorescent images showing the flow of calcium without stimulation in the small molecule (SM)-treated adult canine dermal fibroblasts (ACDFs) on Day 12. (**B**) Graph representing the change in fluorescent intensity over 60 s without stimulation in ACDFs, vehicle-treated control (Ctrl) ACDFs, and SM-treated ACDFs (*n* = 4). (**C**) Fluorescent images before and after 75 s of stimulation with glutamate or KCl. Bars = 50 µm in panel A and 200 µm in panel C. (**D**,**E**) Graph representing the change in fluorescent intensity in SM-treated ACDFs (solid line, *n* = 10), ACDFs (dotted line, *n* = 5) and control (dashed line, *n* = 5) after (**D**) glutamate or (**E**) KCl stimulation. (**F**,**G**) A dot plot graph representing the increased fluorescent intensity 75 s after (**F**) glutamate or (**G**) KCl stimulation. The white dots in the dot plot graph represent the relative fluorescent intensity in each of the cells. Black bars and error bars represent mean and standard deviation, respectively (*n* = 60). * *p* < 0.001.

**Figure 5 ijms-24-15804-f005:**
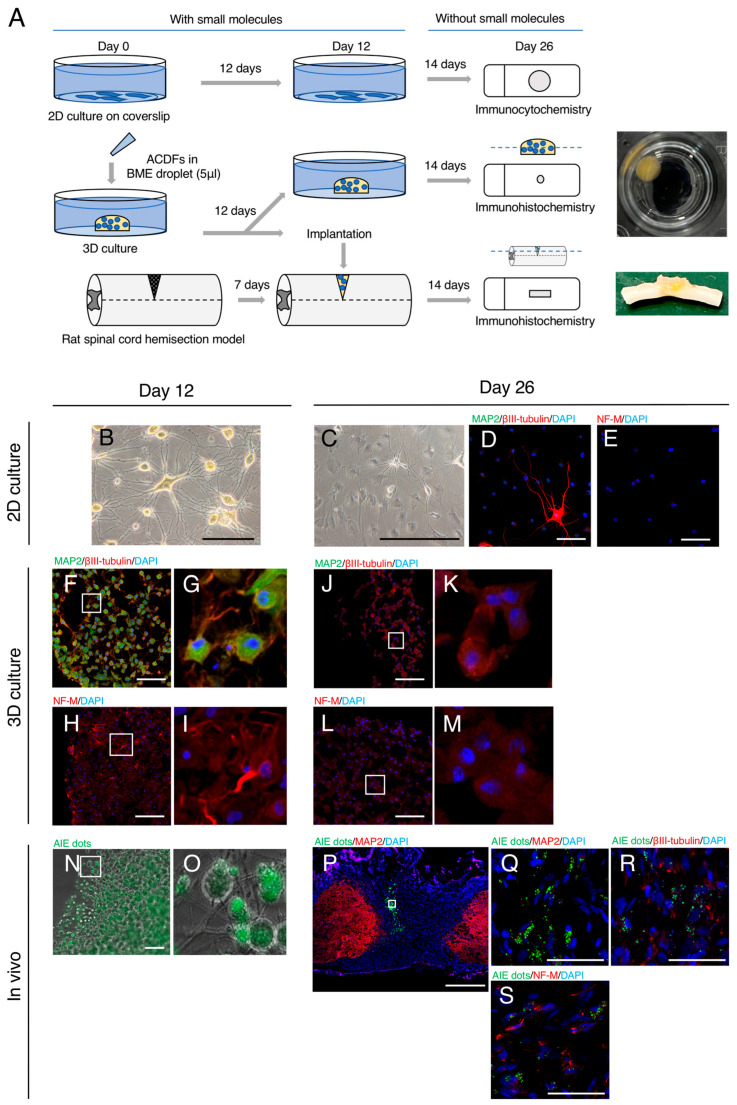
Fate of adult canine dermal fibroblasts (ACDFs) exhibiting neuronal features in 2D culture, 3D-constructed basement membrane extract (BME) droplets, and under in vivo conditions of spinal cord injury. (**A**) Procedures of 2D culture, 3D-constructed BME droplet culture, and in vivo implantation. The photomacrograph represents constructed BME droplets that were cultured for 26 days and an injured spinal cord implanted with BME droplets. (**B**) A photomicrograph of SM-treated ACDFs on Day 12. (**C**) A photomicrograph of SM-treated ACDFs on Day 26 cultured for an additional 14 days without SMs after neuronal induction. (**D**,**E**) A fluorescent image of SM-treated ACDFs on Day 26. (**F**–**I**) Fluorescent images of frozen sections of SM-treated ACDFs in BME droplets on Day 12. (**G**,**I**) High-magnification images of (**F**,**H**), respectively (white square). (**J**–**M**) Fluorescent images of frozen sections of SM-treated ACDFs in BME droplets on Day 26. (**K**,**M**) High-magnification images of (**J**,**L**), respectively (white square). (**N**,**O**) A phase contrast image of aggregation-induced emission (AIE) dot-labeled ACDFs treated with SMs in BME droplets for 12 days. (**O**) A high-magnification image of panel N (white square). (**P**–**S**) Immunohistochemistry of injured spinal cord transplanted AIE dot-labeled ACDFs treated with SMs in BME droplets. (**P**) A low-magnification image of the injured spinal cord on Day 26. (**Q**–**S**) Fluorescent images representing immunoreactivity of cells labeled with AIE dots for (**Q**) MAP2, (**R**) βIII-tubulin, and (**S**) NF-M. Bars = 100 μm in (**B**–**F**,**J**,**H**,**L**,**Q**–**S**), 200 μm in (**N**), and 1000 μm in (**P**).

## Data Availability

The data presented in this study are included in the manuscript and Appendix A and also available on request from the corresponding author.

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
