# Peer review of "Small Molecules Temporarily Induce Neuronal Features in Adult Canine Dermal Fibroblasts"

_ijms, 2023, doi:10.3390/ijms242115804_

Round 1
Reviewer 1 Report
Comments and Suggestions for Authors
IJMS2650351 comments
Somatic cells, including fibroblasts and astrocytes, can be directly converted into neuronal cells using either transcriptional factors or small chemical molecules (SMs). In this manuscript, Arai et al used a cocktail of SMs to induce adult canine dermal fibroblasts (ACDFs) into neuronal-like cells. They found that SMs induced neuronal morphology and features in ACDFs. After characterization of induced cells by immunochemical staining, microarray, 3D culture, and in vivo spinal implantation in the injury microenvironment, they showed that the induction of neuronal features in ACDFs was temporary.
This study used the reported SM cocktail to induce fibroblasts (ACDFs) into neuronal-like cells in canine. I would suggest to meet the novelty criteria in IJMS by addressing the following points.
1. Several groups have reported the successful induction of human and mouse fibroblasts into functional neurons using SMs. The authors may consider optimizing their protocol or screen new SMs to induce functional neurons, instead of the “temporary features” in ACDFs. For example, P7C3-A20 and ISX9 have been shown as the two most significant components in the neuronal reprogramming (Yang Y, Stem Cell Reports. 2019;13:862, PMID: 31631018).
2. The neuronal reprogramming process using SMs sometimes is very tricky and needs special skills. To exclude the involvement of technical issues, the authors may need use human or mouse fibroblasts as a side-by-side positive control.
3. As an alternative strategy for neuronal regeneration, some neuronal functional assays are needed to test the therapeutic potential of these SM-induced cells.
4. Further characterization of epigenetic and chromatin remodeling of SM-induced cells may provide mechanistic clue why these SMs cannot induce permeant functional neurons in canine fibroblasts as did in human and mouse fibroblasts.
5. The authors may discuss from the point of regenerative medicine why the canine model, instead of the pig, is selected for this study.
Author Response
Thank you very much for taking the time to review this manuscript. We have revised the manuscript according to your comments. Please find the detailed responses below and the corresponding revisions in the re-submitted files.
- Several groups have reported the successful induction of human and mouse fibroblasts into functional neurons using SMs. The authors may consider optimizing their protocol or screen new SMs to induce functional neurons, instead of the “temporary features” in ACDFs. For example, P7C3-A20 and ISX9 have been shown as the two most significant components in the neuronal reprogramming (Yang Y, Stem Cell Reports. 2019;13:862, PMID: 31631018).
Response: Thank you very much for your kind and beneficial advice. As you indicated, further screening of SMs is needed for optimizing our protocol. We plan to conduct a research study focused on SM screening that would include the two SMs proposed by you. In order to conduct efficient screening, it will be necessary to clarify the induction mechanisms and reprogramming barriers through comprehensive analysis.
This point was addressed in the Discussion section as follows: (L276–282)
In this study, when the neuronal induction process using the eight SMs was completed on Day 6, the proportion of ACDFs showing neuronal morphology and expressing βIII-tubulin were 85% and 49%, respectively. These results indicate that some of ACDFs are resistant to SM-induced neuronal conversion. Further optimization of the induction process is necessary to achieve highly efficient conversion of ACDFs to neurons. This requires efficient and effective SM screening based on the neuronal induction mechanisms and reprogramming barriers.
- The neuronal reprogramming process using SMs sometimes is very tricky and needs special skills. To exclude the involvement of technical issues, the authors may need use human or mouse fibroblasts as a side-by-side positive control.
Response: Thank you very much for your careful advice. To exclude technical issues, we conducted individual assays using nine fibroblast lines isolated from different dogs. Since we used primary ACDFs, there is some variation in the efficiency of induction of neuronal features (Figure 1M and Figure 2I), but generally consistent neuronal inducibility was obtained.
In particular, morphological changes of our ACDFs were more pronounced than those of human fibroblasts in previous studies [Hu, W. Cell Stem Cell 2015, 17, 204–212, Figure 1B, Yang, Y. Stem Cell Reports 2019, 13, 862–876, Figure 1D]. In addition, the expression level of neuronal protein was increased to a similar level in human fibroblasts [Hu, W. Cell Stem Cell 2015, 17, 204–212 Figure S1B and C, Yang, Y. Stem Cell Reports 2019, 13, 862–876, Figure 2M and N]. We considered neuronal induction partially successful until additional cultures were utilized and in vivo transplantation was performed. While it is important to use cells of other species as controls, this method has not been established in other species, nor has it been fully tested especially in terms of permanency of the induced neuronal features.
We have proposed as follows: (L315–317)
It is important to confirm the permanency of the induced neuronal features without induction factors, in order to confirm that the neuronal conversion was completely successful.
As you noted in your 4th comment, we considered that the induced neuronal features were temporary due to epigenetic barriers.
The information about epigenetic barriers is as follows: (L320–324)
These studies imply that incomplete neuronal induction by SMs, which fail to fully reprogram the original epigenetic memory of ACDFs, result in a reversion to the original phenotype. A comprehensive epigenetic analysis and screening for SMs that can effectively promote epigenetic reprogramming are necessary to completely generate permanently converted neurons from ACDFs.
- As an alternative strategy for neuronal regeneration, some neuronal functional assays are needed to test the therapeutic potential of these SM-induced cells.
Response: Thank you very much for your recommendation. In order to use the neuronal induced cells for regenerative medicine, it is necessary to prove in vitro and in vivo that the neuronal induced cells possess neuronal function.
This description was added as follows: (L337–341)
In addition, in order to use the neuronal induced cells for regenerative medicine, it is necessary to prove in vitro and in vivo that the cells have neuronal function to test the therapeutic potential. In this study, we have shown that calcium ion influx occurs in SM-treated ACDFs spontaneously and in response to glutamate or KCl stimulation. However, additional electrophysiological evaluation will more reliably assess neuronal function.
- Further characterization of epigenetic and chromatin remodeling of SM-induced cellsmay provide mechanistic clue why these SMs cannot induce permeant functional neurons in canine fibroblasts as did in human and mouse fibroblasts.
Response: Thank you very much for your precise advice. I completely agree with you. Similar to our study, it is known that canine iPSC generation does not work efficiently using the protocol for human cells due to the expected epigenetic barriers (Kuzma-Hunt AG, et al. Stem Cells Dev. 2023. PMID: 36884307). Clarifying the epigenetic barriers is the key to success in neuronal induction using SM. The discussion about epigenetic barrier is included in L320–324.
These studies imply that incomplete neuronal induction by SMs, which fail to fully reprogram the original epigenetic memory of ACDFs, result in a reversion to the original phenotype. A comprehensive epigenetic analysis and screening for SMs that can effectively promote epigenetic reprogramming are necessary to completely generate permanently converted neurons from ACDFs.
- The authors may discuss from the point of regenerative medicine why the canine model, instead of the pig, is selected for this study.
Response: Thank you very much for your recommendation.
The description about this was added as follows: (L342–349)
When conducting research in the anticipation of future applications in regenerative medicine, it may be appropriate to use canine rather than other species. Canines share not only the same physiological functions and aging phenomena, but also the same living environment as humans. Therefore, the two species share many common diseases and pathogenic processes. For instance, brain trauma and spinal cord injury in canines, like in humans, occur in traffic accidents and fall accidents. It is expected that clinical research and veterinary clinical trials using canine cells will be useful as reliable preclinical data to initiate clinical trials on humans.
Reviewer 2 Report
Comments and Suggestions for Authors
This manuscript describes the neuronal differentiation of adult canine dermal fibroblasts (ACDFs) by small molecules (SMs) using previously established protocols for human and rodent cells. The methods are well described. The results are clear, and the conclusions drawn are mostly agreeable. Although short, the discussion is adequate. The only weakness is the interpretation of some of the data. The authors sometimes overlook the statistical analyses associated with the data. This manuscript should be accepted after minor revision. Specific comments are the following:
L81-82 This sentence is not an accurate description of the data presented in Fig. S2B. There are no statistically significant differences indicated among VCRFYS, VCRFYSG, and VCRFYSGD. The employment of VCRFYSGD over 6 or 7 SMs should be more precisely discussed.
L84-86 This text should be improved to better describe the data. For example, the addition of b-FGF together with SMs increased the proportion of βIII-tubulin positive cells and decreased that of dead cells. However, the further addition of three other GFs did not improve induction efficiency or cell death.
L88 “studyhn” is this a typo? This should be fixed.
L95-97 Is the increase from Day to Day 6 or to Day 12 statistically significant? If not, this needs to be changed. Fig. 1M seems to indicate there are no significant differences among SM-D1, SM-D6, and SM-D12.
L136 “untreated” It is not clear how the untreated cells were prepared. Were these cells cultured in the same way as control and SM-treated cells without vehicle or SMs? Untreated cells need some descriptions in the method section.
Fig. 3GH These data seem to be qRT-PCR data. If so, the legend should include this information.
L249 “for to induce” is this a typo? This should be fixed.
L472-473 The last sentence does not make sense. Please fix this.
L493 “(“ is missing.
Comments on the Quality of English LanguageEnglish is adequate.
Author Response
Thank you very much for taking the time to review this manuscript. We have revised the manuscript according to your comments. Please find the detailed responses below and the corresponding revisions in the re-submitted files.
- L81-82 This sentence is not an accurate description of the data presented in Fig. S2B. There are no statistically significant differences indicated among VCRFYS, VCRFYSG, and VCRFYSGD. The employment of VCRFYSGD over 6 or 7 SMs should be more precisely discussed.
Response: Thank you very much for your observation and subsequent recommendation.
The following changes have been made in the manuscript: (L80–88)
Addition of Go6983, a protein kinase C (PKC) inhibitor to the six SMs did not increase the proportion of neuronal morphology. However, according to reports that state Go6983 increases the expression of neuronal transcriptional factors in human fibroblasts [6], we included Go6983 in our protocol to upregulate neuronal genes. Next, we added Dorsomorphin, one of the bone morphogenetic protein (BMP) type I receptor inhibitors that are used in many protocols and contribute to induction of neuronal features in human and rat fibroblasts [5,7,8,12]. As a result, Dorsomorphin increased the proportion of neuronal morphology by 5% as compared to the seven SMs (Figures S2A and B), which qualified Dorsomorphin for inclusion in our protocol.
- L84-86 This text should be improved to better describe the data. For example, the addition of b-FGF together with SMs increased the proportion of βIII-tubulin positive cells and decreased that of dead cells. However, the further addition of three other GFs did not improve induction efficiency or cell death.
Response: Thank you very much for your kind advice.
We corrected the description as follows: (L90–95)
The addition of b-FGF with SMs increased the proportion of βIII-tubulin–positive cells and decreased nuclear fragmented dead cells (Figure S3B and C). However, the further addition of neurotrophins did not improve induction efficiency and inhibit cell death. According to these results of preliminary study, ACDFs were cultured for 6 days in a neuronal induction medium containing eight SMs and b-FGF in the principal experiments.
- L88 “studyhn” is this a typo? This should be fixed.
Response: Thank you very much for your point out. We corrected “studyhn” to “study.”
- L95-97 Is the increase from Day to Day 6 or to Day 12 statistically significant? If not, this needs to be changed. Fig. 1M seems to indicate there are no significant differences among SM-D1, SM-D6, and SM-D12.
Response: Thank you very much for your precise advice. We detected no significant differences between SM-D1, SM-D6, and SM-D12.
The description was changed as follows: (L102–107)
The proportions of ACDFs exhibiting neuronal morphology (Figure 1L) on Day 1 (80.9%), Day 6 (85.2%), and Day 12 (89.1%) were significantly increased compared to control cells across all time points (Figure 1M). Although there were not statistically significant, the proportions of ACDFs exhibiting neuronal morphology showed an increasing trend from Day 1 to Day 12.
- L136 “untreated” It is not clear how the untreated cells were prepared. Were these cells cultured in the same way as control and SM-treated cells without vehicle or SMs? Untreated cells need some descriptions in the method section.
Response: Thank you very much for your advice and pointing out of lacunae in our study. The representation, “untreated cell” means ACDF before receiving treatments. All the representation, “untreated cell” in manuscript were changed to “ACDF.”
In addition, the description in the results section has been slightly modified as follows: (L140–152)
Next, for comprehensive analysis, we compared global gene expression profiles between ACDFs before and after 12-day SMs treatment using microarray. Hierarchical clustering analysis revealed similar genome-wide changes between ACDFs before and after 12-day SMs treatment (Figure 3A). A total of 15,170 genes were analyzed without duplication, where 1,588 and 1,456 genes were significantly upregulated and downregulated, respectively in SM-treated ACDFs as compared with ACDFs before SMs treatment. Among the top 100 upregulated genes, 13 genes highly associated with neurons, such as SNAP25, KCNA1, and GRIA4, were identified (Figure 3B). Furthermore, among the top 100 downregulated genes, 10 genes highly associated with fibroblasts, such as COL12A1, SVEP1, and CCN5, were identified (Figure 3C). The gene ontology (GO) and pathway analyses revealed that neuron-related terms such as brain development, neuron projection development, and synaptic vesicle cycle were upregulated in SM-treated ACDFs as compared with ACDFs before SMs treatment (Figure 3D).
- Fig. 3GH These data seem to be qRT-PCR data. If so, the legend should include this information.
Response: Thank you very much for pointing this out.
We added information of qRT-PCR in Fig. 3 legend as follows: (L170–171)
Time-course analysis of relative gene expression level of (G) MAP2 and (H) TUBB3 using quantitative real-time reverse transcription-polymerase chain reaction.
The abbreviation “qRT-PCR” has been extended in Results section as follows: (L157–158)
quantitative real-time reverse transcription-polymerase chain reaction (qRT-PCR)
- L249 “for to induce” is this a typo? This should be fixed.
Response: Thank you very much for your keen observation. We have deleted “for” from the sentence.
- L472-473 The last sentence does not make sense. Please fix this.
Response: Thank you very much for your advice.
This sentence has been changed as follows: (L501–502)
The immunostaining procedures were performed as per the same protocol for immunocytochemistry.
(L532–533) The immunohistochemical procedures were performed using the same protocol as immunocytochemistry.
- L493 “(“ is missing.
Response: Thank you very much for pointing this out.
We added “(“ in the sentence as follows: (L522)
AIE dots (LuminiCell Tracker 540-Cell Labeling Kit, Sigma-Aldrich)
Round 2
Reviewer 1 Report
Comments and Suggestions for Authors
I have no more questions